# Constructing a Prognostic Model for Clear Cell Renal Cell Carcinoma Based on Glycosyltransferase Gene and Verification of Key Gene Identification

**DOI:** 10.3390/ijms262010182

**Published:** 2025-10-20

**Authors:** Chong Zhou, Mingzhe Zhou, Yuzhou Luo, Ruohan Jiang, Yushu Hu, Meiqi Zhao, Xu Yan, Shan Xiao, Mengjie Xue, Mengwei Wang, Ping Jiang, Yunzhen Zhou, Xien Huang, Donglin Sun, Chunlong Zhang, Yan Jin, Nan Wu

**Affiliations:** 1Laboratory of Medical Genetics, Harbin Medical University, Harbin 150081, China; 202201034@hrbmu.edu.cn (C.Z.); zhoumingzhe0228@163.com (M.Z.); luoyuzhou@hrbmu.edu.cn (Y.L.); jiangruohan_sdfmu@163.com (R.J.); 2020173018@hrbmu.edu.cn (Y.H.); 201901010@hrbmu.edu.cn (M.Z.); 15774511409@163.com (X.Y.); xiaoshan970513@163.com (S.X.); xuemengjie2000@126.com (M.X.); wmw200102@163.com (M.W.); jjennifer_0417@163.com (P.J.); zhouyunzhen@hrbmu.edu.cn (Y.Z.); 202301043@hrbmu.edu.cn (X.H.); sundl@hrbmu.edu.cn (D.S.); 2Key Laboratory of Preservation of Human Genetics Resources and Disease Control in China (Harbin Medical University), Ministry of Education, Harbin 150081, China; 3College of Bioinformatics Science and Technology, Harbin Medical University, Harbin 150081, China; zhangchunlong@hrbmu.edu.cn; 4State Key Laboratory of Frigid Zone Cardiovascular Diseases, Harbin Medical University, Harbin 150081, China

**Keywords:** clear cell renal cell carcinoma, glycosyltransferase, risk model, machine learning algorithms, TYMP, GCNT4

## Abstract

Clear cell renal cell carcinoma (ccRCC) is the most common and aggressive subtype of kidney cancer. This study aimed to construct a prognostic model for ccRCC based on glycosyltransferase genes, which play important roles in cell processes like proliferation, apoptosis. Glycosyltransferase genes were collected from four public databases and analyzed using RNA-seq data with clinical information from three ccRCC datasets. Prognostic models were constructed using eight machine learning algorithms, generating a total of 117 combinatorial algorithm models, and the StepCox[forward]+Ridge model with the highest predictive accuracy (C-index = 0.753) which selected and named the Glycosyltransferases Risk Score (GTRS) model. The GTRS effectively stratified patients into high- and low-risk groups with significantly different overall survival and maintained robust performance across TCGA, CPTAC, and E-MTAB1980 cohorts (AUC > 0.75). High-risk patients exhibited higher tumor mutational burden, immunosuppressive microenvironment, and poorer response to immunotherapy. TYMP and GCNT4 were experimentally validated as key genes, functioning as oncogenic and tumor-suppressive factors. In conclusion, GTRS serves as a reliable prognostic tool for ccRCC and provides mechanistic insights into glycosylation-related tumor progression.

## 1. Introduction

Clear cell renal cell carcinoma (ccRCC) is the most common and aggressive subtype of kidney cancer, making up about 75% of cases [1]. Around 330,000 new cases are diagnosed worldwide each year, and this number is rising [2]. Although advancements in diagnostic techniques and therapeutic strategies have contributed to a decline in mortality rates in recent years, challenges remain. Postoperative recurrence, metastasis, and therapeutic resistance remain obstacles in the management of advanced disease, substantially limiting prognostic outcomes [3]. At present, there is a lack of accurate indicators for predicting the outcome of ccRCC. Furthermore, reliance on single molecular biomarkers is insufficient for assessing tumor complex progression. Therefore, establishing a multifactorial prognostic model is necessary.

Glycosyltransferases are key catalytic enzymes that mediate glycosylation, a widespread post-translational modification involved in regulating protein and lipid functions [4]. Aberrant expression or activity of glycosyltransferases can lead to altered glycosylation patterns, which are closely associated with tumor development and immune evasion [5]. For instance, N-glycosylation of PD-L1 has been shown to stabilize its expression and contribute to immunotherapy resistance, while inhibition of specific glycosyltransferases can enhance anti-tumor immunity [6,7]. Aiming at this mechanism, inhibiting the catalytic subunit A of the oligosaccharyltransferase complex can significantly reduce the glycosylation level of PD-L1 and enhance the efficacy of anti-PD-1 therapy [8]. The critical role of glycosylation-related genes, particularly glycosyltransferases, has prompted the development of prognostic models based on glycosyltransferase gene sets in various cancers [9]. For example, knockout of the alpha-1,3-mannosyltransferase (ALG3) gene can lead to a decrease in the glycosylation level of PD-L1, reducing its binding to PD-1, thereby enhancing the killing effect of T cells on tumors [10].

Dysregulated glycosylation has been implicated in ccRCC progression, angiogenesis, and immune modulation [3]. Several studies have reported that altered expression of specific glycosyltransferases, such as FUT3 and MGAT5, promotes epithelial–mesenchymal transition, enhances VEGF-mediated angiogenesis, and facilitates immune evasion by modulating PD-L1 stability and immune cell infiltration in ccRCC [11,12,13]. Although such models have been established in several tumor types, their exploration in ccRCC remains limited. It is necessary to construct a robust glycosyltransferase-based gene signature for ccRCC prognosis, leveraging comprehensive bioinformatics analysis to improve predictive accuracy and provide novel insights into the glycan-related mechanisms underlying ccRCC progression.

Risk prediction modeling has shifted toward machine learning algorithms, particularly in oncology prognosis, and advancements in algorithmic techniques have significantly enhanced predictive accuracy and clinical utility. With the evolution of machine learning, methods such as support vector machines and random forests are now widely employed for feature selection and classification. For example, random forest algorithms can achieve an accuracy of 88.89% in discriminating between benign and malignant pancreatic cystic lesions based on radiomic features, with their ensemble learning approach effectively mitigating overfitting issues inherent in single decision trees [14]. Furthermore, risk prediction models show considerable promise in prognostic stratification and predicting therapeutic sensitivity in cancer [15,16]. Thus, integrating molecular markers with clinical characteristics within prognostic risk scoring systems can substantially improve risk stratification accuracy in patients with ccRCC.

This study aims to construct a prognostic model for ccRCC based on glycosyltransferase genes, reveal the prognostic value of glycosyltransferases in ccRCC, and further identify and validate key genes among them. Figure 1 demonstrates the main content of this research. 

## 2. Results

### 2.1. Identification of Prognosis-Associated Differentially Expressed Glycosyltransferase Genes in ccRCC

A total of 337 glycosyltransferase genes were initially collected from GGDB, MsigDB, HGNC, and Glycosmos databases (Appendix A). To identify glycosyltransferase genes which are differentially expressed in ccRCC and associated with patient prognosis, we performed differential gene expression analysis on the TCGA dataset using two distinct algorithms: DESeq2 and edgeR (Figure 2A,B). The overlap of these significantly differentially expressed genes identified by both algorithms with the curated list of 337 glycosyltransferase genes produced 55 differentially expressed glycosyltransferase genes (Figure 2C). Subsequently, these 55 genes were subjected to univariate Cox proportional hazards regression analysis to screen for those significantly associated with ccRCC prognosis. This analysis identified 16 prognostic glycosyltransferase genes (Figure 2D). Among these, 4 genes were classified as risk genes, and 12 genes were classified as protective genes. Heatmap analysis further revealed that the 4 risk genes were highly expressed in cancer samples, whereas the 12 protective genes showed higher expression levels in normal tissues, consistent with their prognostic classification (Figure 2E).

### 2.2. Construction and Validation of the Glycosyltransferase-Related Signature (GTRS) Prognostic Model for ccRCC

Three independent ccRCC transcriptomic datasets with prognostic data were used to evaluate the impact of glycosyltransferase on the prognosis of ccRCC: TCGA-KIRC dataset (*n* = 437) as a training set and EMTAB1980 dataset (*n* =101) and CPTAC-ccRCC dataset (*n* = 103) as two validation sets.

The 16 prognostic GT genes identified previously were incorporated into the model-building process. We integrated 8 distinct machine learning algorithms, generating a total of 117 combinatorial algorithm models to construct the prognostic signature (Figure 3A). Among the hundreds of models built using this comprehensive machine learning approach, the Stepcox[forward] + Ridge algorithm had the highest mean Concordance index (C-index) of 0.753.

The median risk score calculated by the Stepcox[forward] + Ridge model divided ccRCC patients into high- and low-risk groups. Kaplan–Meier survival curves showed that among all cohorts, the overall survival (OS) times of high-risk group patients is significantly shorter than that of low-risk group (Figure 3B–D). Subsequently, time-dependent Receiver Operating Characteristic (ROC) curve analysis was performed to evaluate the predictive accuracy of the Stepcox[forward] + Ridge model. The AUC values of the TCGA cohort were 0.824 (1-year), 0.763 (3-year), and 0.765 (5-year). The AUC values of the EMTAB1980 cohort were 0.819 (1-year), 0.823 (3-year), and 0.783 (5-year). The AUC values of the CPTAC cohort were 0.816 (1-year), 0.754 (3-year), and 0.731 (5-year) (Figure 3E–G). These results indicate that the Stepcox[forward] + Ridge model demonstrates good performance in predicting patient outcomes.

Based on the evaluation results above, the StepCox[forward] + Ridge algorithm was selected to construct the prognostic model based on 13 glycosyltransferase genes and named the Glycosyltransferases Risk Score (GTRS). The risk score formula is defined as follows: Risk Score = (0.184081692 × TYMP expression) + (0.091260892 × B4GALNT4 expression) + (0.090327624 × PLOD2 expression) + (0.018179472 × CDS1 expression) + (0.010026151 × GALNT14 expression) + (−0.014497718 × HS6ST1 expression) + (−0.055940121 × CHST9 expression) + (−0.072841937 × ST6GAL1 expression) + (−0.102573603 × HS3ST2 expression) + (−0.131885590 × UGT2B7 expression) + (−0.180896639 × GCNT4 expression) + (−0.187697409 × FUT3 expression) + (−0.226785547 × UGT8 expression).

### 2.3. Prognostic Risk Score GTRS as an Independent Prognostic Factor for ccRCC Patients

To determine whether the GTRS model could serve as an independent prognostic factor for ccRCC, we performed univariate Cox regression and multivariate regression analyses to screen for factors with independent prognostic value associated with OS in the TCGA ccRCC cohort. The variables included in the analysis encompassed the risk score from the GTRS model, along with clinically relevant factors such as age, race, sex, pathological grade, and clinical stage. The univariate Cox regression results revealed that GTRS, age, pathological grade, and clinical stage were associated with patient prognosis. High GTRS (HR = 5.10, 95% CI = 3.75–6.93), age ≥ 64 years (HR = 1.77, 95% CI = 1.27–2.47), pathological grade G3 and G4 (HR = 2.73, 95% CI = 1.85–4.04), and clinical stage III and IV disease (HR = 4.52, 95% CI = 3.11–6.56) were significantly associated with increased risk of death (Appendix A). Multivariate Cox regression analysis demonstrated that GTRS (HR = 3.61, 95% CI = 2.58–5.04), age (HR = 1.58, 95% CI = 1.13–2.22), pathological grade (HR = 3.21, 95% CI = 2.15–4.77), and clinical stage (HR = 1.55, 95% CI = 1.02–2.35) were all independent predictors of patient OS (Appendix A).

To further validate GTRS evaluation effect, three independent cohorts were integrated into a Meta-cohort after batch effect correction (Appendix A), and the GTRS model was calculated for this Meta-cohort. Both univariate and multivariate Cox regression analyses within the Meta-cohort confirmed that GTRS, age, pathological grade, and clinical stage were independent prognostic factors for patient OS. (Appendix A). These results indicate that the GTRS, patient age, pathological grade, and tumor stage are all independent prognostic factors for overall survival in ccRCC patients.

Next, univariate Cox regression analysis was employed in the three cohorts to validate the predictive ability of the model for survival outcomes. The hazard ratios (HRs) of the TCGA, EMTAB1980, and CPTAC cohorts were 3.94, 4.22, and 3.15. We also assessed a meta-analysis of Combined Cohorts which contain data from the three cohorts above (Figure 4A). These results collectively indicate that the prognostic model constructed using the Stepcox[forward]+Ridge algorithm effectively predicts survival outcomes in all three independent ccRCC cohorts.

To enhance predictive capability, a nomogram incorporating the independent prognostic factors was further developed based on the TCGA ccRCC (Figure 4B). Calibration curve analysis demonstrated that the predicted curves for 1-year, 3-year, and 5-year survival closely aligned with the actual curves, with the 5-year survival prediction exhibiting the best performance (Figure 4C). The nomogram achieved the highest C-index value. While the GTRS ranked second to the nomogram, it outperformed other individual indicators. Decision Curve Analysis (DCA) results indicated that the nomogram provided significantly higher net benefit than other models across the risk threshold range of 0.00–1.00. (Figure 4D). The “All” strategy showed higher net benefit at low-risk thresholds but declined sharply as the threshold increased, eventually falling below the “None” strategy when the threshold exceeded 0.55 (Figure 4E). Comparison among the models demonstrated that the nomogram maintained the optimal net benefit over a wide risk threshold range, showcasing superior clinical predictive utility.

The prognostic risk score GTRS was compared with 27 other prognostic models randomly selected from other studies in terms of C-index. A heatmap was used to visualize the hazard ratios and significance levels of GTRS and other models across the TCGA, EMTAB1980, and CPTAC datasets (Figure 5A). GTRS demonstrated the third-best performance after the models developed by the Ren and Yang teams, showing higher hazard ratios and significance across the three cohorts. Further analysis of the C-index of these models revealed that in the TCGA cohort, GTRS ranked behind the models by Ren, Wu, and Yang, but achieved the highest C-index in the CPTAC cohort and outperformed 20 other models in the EMTAB1980 cohort (Figure 5B). Comparative analysis of the area under the ROC curve (AUC) at 1-, 3-, and 5-year follow-ups showed that GTRS exhibited high stability in predicting 1-year AUC values and led in the TCGA cohort (Figure 5C–E). These results indicate that the GTRS model has strong predictive performance among similar studies.

### 2.4. Comprehensive Analysis of GTRS in Relation to Tumor Mutational Burden and Immune Infilation

Tumor mutational burden (TMB) is closely associated with tumor aggressiveness and the efficacy of targeted therapy and immunotherapy. We further analyzed TMB within the high- and low-risk GTRS groups using TCGA mutation data. The top 20 most frequently mutated genes in each group were visualized. The results showed that in the high-risk group, the top five mutated genes and their mutation frequencies were PBRM1 (43%), VHL (41%), TTN (24%), SETD2 (22%), and BAP1 (15%) (Figure 6A). In the low-risk group, the top five mutated genes were VHL (51%), PBRM1 (43%), TTN (12%), MTOR (9%), and BAP1 (9%) (Figure 6B).

To evaluate the immune infiltration status and tumor microenvironment characteristics in the high- and low-risk GTRS groups, the ESTIMATE algorithm was employed for a comprehensive assessment of the tumor microenvironment. Results demonstrated that the high-risk group exhibited significantly higher ESTIMATE scores and Immune Scores (Figure 6C). To analyze immune cell types, the CIBERSORT and ssGSEA algorithms were employed. Results revealed significantly higher infiltration of regulatory T cells, M0 macrophages, neutrophils, myeloid-derived suppressor cells, monocytes, and Th2/Th17 cells in the high-risk group (Figure 6D,E). This immunosuppressive cellular profile may contribute to poorer prognosis. Further analysis using the TIDE algorithm revealed significantly higher scores for TIDE, dysfunction, exclusion, Myeloid-Derived Suppressor Cells (MDSC), and Cancer-Associated Fibroblasts (CAFs) in the high-risk group compared to the low-risk group (Figure 7A). These results suggest a poorer response to immunotherapy and potentially adverse prognosis in high-risk patients. Additionally, immune checkpoint genes PDCD1, CTLA4, LAG3, and LGALS9 exhibited elevated expression in the high-risk group, indicating a possible state of T cell exhaustion (Figure 7B).

To analyze differences in drug sensitivity between GTRS risk groups and screen for potential therapeutic agents, we assessed drug responses in high- and low-risk GTRS groups using GDSC data. Among the top 10 compounds meeting the criteria of average IC50 < 5 and smallest *p*-values, significantly lower mean IC50 values were observed in the low-risk group compared to the high-risk group (Figure 7C).

These findings demonstrate that the high-risk GTRS group exhibits a more aggressive mutational profile and immunosuppressive tumor microenvironment, indicating potential resistance to immunotherapy. Thus, the GTRS scoring system holds dual utility for prognostic stratification and therapeutic guidance.

### 2.5. Screening and Analysis of Key Genes in Prognostic Models

During construction of the GTRS risk model, genes selected by diverse integrated machine learning algorithms were statistically analyzed. The results demonstrated that TYMP, GCNT4, UGT2B7, HS3ST2, and FUT3 were consistently selected by all 17 valid algorithms and served as key factors in every model iteration (Figure 8A,B). Transcriptome levels of these five pivotal genes were subsequently compared between TCGA tumor samples and GTEx healthy kidney tissues. Analytical results revealed TYMP and HS3ST2 in tumor samples were significantly upregulated and GCNT4 and FUT3 were significantly downregulated (Figure 8C). We noticed the expression levels of UGT2B7 were not significantly different, so we excluded UGT2B7 from subsequent analyses. Further survival analyses demonstrated that high expression of TYMP was associated with shorter overall survival. In contrast, high expression of GCNT4, HS3ST2, and FUT3 was significantly correlated with longer overall survival (Figure 8D). Similarly, analysis of disease-free survival revealed that elevated expression of GCNT4 and FUT3 was associated with a longer DFS, further supporting their potential protective roles in ccRCC progression (Appendix A).

UALCAN was utilized to analyze the protein expression of risk factors in CPTAC ccRCC samples; we found that only TYMP exhibited a detectable upregulated protein level in ccRCC patients (Appendix A). Further analysis using HPA database revealed strong positive staining of TYMP in ccRCC tissues. In contrast, GCNT4 expression was detected in normal kidney tissues; no observable staining was found in ccRCC tissues (Appendix A). These findings indicate that TYMP is significantly highly expressed, and GCNT4 is significantly downregulated in ccRCC. Due to the lack of corresponding protein expression data for HS3ST2 and FUT3, TYMP and GCNT4 were selected for further analysis.

Univariate Cox regression analysis of TYMP found that its high expression was significantly associated with poor patient prognosis. Multivariate Cox regression analysis incorporating TYMP and age, gender, stage, and purity also found HR bigger than 1 for TYMP, indicating TYMP as an independent risk factor for unfavorable outcomes (Appendix A). In contrast, GCNT4 was identified as an independent protective prognostic factor (Appendix A).

### 2.6. Functional Validation of TYMP and GCNT4 in ccRCC

We further investigate the function of TYMP and GCNT4 in ccRCC through in vitro experiments. According to the endogenous TYMP and GCNT4 in four types of ccRCC cells, A498 was used to establish a stable knockdown model for TYMP, and HK-2 was used to construct a GCNT4 overexpressing cell line (Figure 9A–C). CCK-8 assay demonstrated that knockdown of TYMP decreased the cell proliferation ability, and the overexpression of GCNT-4 also decreased the cell proliferation ability (Figure 9D,E). We used the Tunnel assay to detect cell apoptosis. The knockdown of TYMP increased cell apoptosis and overexpressed the expression of GCNT4, which also increased cell apoptosis (Figure 9F,G). Migration and invasion assays demonstrated that knockdown of TYMP decreased the cell motility, and overexpression of GCNT4 also decreased the motility (Figure 9H,I). The wound healing assay results are consistent with the migration and invasion assays, showing that knockdown of TYMP and overexpression of GCNT4 significantly impaired cell migration (Figure 9J,K). These results suggest that TYMP is associated with poor prognosis and functions as an oncogene by promoting malignant behaviors in ccRCC, whereas GCNT4, linked to favorable clinical outcomes, exhibits tumor-suppressive roles by inhibiting proliferation, migration, and invasion.

## 3. Discussion

Aberrant protein glycosylation is a hallmark of cancer and critically implicated in tumor invasion, metastasis, drug resistance, and immune evasion [17,18]. Glycosyltransferases are the direct executors of glycosylation and drive tumor progression when dysregulated. Notably, increased evidence suggests that glycosyltransferase-related pathways are closely intertwined with canonical ccRCC signaling networks, including the VHL–HIF axis. Loss of functional VHL impairs the clearance of misprocessed glycoproteins and increases cellular sensitivity to glycosylation stress, indicating a role of pVHL in maintaining protein homeostasis under metabolic stress [19]. Moreover, elevated O-GlcNAcylation, catalyzed by O-GlcNAc transferase, stabilizes HIF-1α and promotes glycolysis through GLUT1, whereas reducing O-GlcNAcylation enhances pVHL-mediated HIF-1α degradation [20]. Similarly, the sialyltransferase ST6Gal-I facilitates HIF-1α accumulation and transcriptional activation under hypoxia, enhancing the expression of glucose metabolism-related genes [21]. These findings suggest that dysregulated glycosyltransferases not only mediate aberrant glycosylation but also modulate the VHL–HIF axis, linking metabolic reprogramming with hypoxia adaptation in tumor progression. While single-gene biomarkers lack robustness for prognostication in heterogeneous tumors, multi-gene models offer superior predictive power and clinical relevance. Despite the established role of glycosyltransferases in tumorigenesis, systematic investigation of their prognostic significance in ccRCC remains lacking.

To systematically investigate the prognostic relevance of glycosyltransferases in ccRCC, we compiled a comprehensive set of 337 glycosyltransferases genes, curated from multiple published sources and four major databases [22,23,24]. Functional enrichment analysis revealed significant associations of these glycosyltransferases with alterations in cell signaling, adhesion, and immune recognition. This supports glycosyltransferase’s modulate tumorigenesis and progression by modifying specific proteins within these critical pathways.

Due to excessive gene inclusion, which may capture noise rather than true biological signals and compromise predictive power and generalizability, we implemented rigorous feature selection to mitigate overfitting risks in prognostic model development. We performed initial gene filtering to eliminate irrelevant features, reduce dimensionality, and enhance model robustness. To minimize false positives, we employed two algorithms to analysis differential expression of glycosyltransferase [25,26]. Subsequently, univariate Cox regression identified 16 prognosis-associated glycosyltransferase genes to the constructed prognosis model. This analytical strategy enhances model stability and generalizability through dimensionality reduction.

Multiple machine learning algorithms, such as LASSO, Ridge, SVM, and Random Forest, can be used for prognostic modeling, each with its own advantages and limitations [27]. We employed 117 modeling combinations derived from eight base algorithms to identify glycosyltransferase risk factors and evaluate their impact on patient survival. This integrative approach enhances predictive accuracy while mitigating overfitting and underfitting [28]. Model performance was assessed using the C-index [29]. The Stepcox[forward] + Ridge algorithm, achieving the highest C-index in both training and validation sets, was selected to construct the GTRS. Comparative analyses of ROC curves, AUC values, and C-index across training and validation sets demonstrated GTRS’s superior risk stratification and predictive performance.

Tumor mutational burden reflects immunogenicity and serves as a key biomarker for predicting immune checkpoint inhibitor efficacy by quantifying somatic nonsynonymous mutations in the tumor genome [24]. Our results demonstrated significantly elevated TMB in the high-risk GTRS group. The tumor microenvironment comprises tumor cells, immune cells, stromal components, vasculature, extracellular matrix, and signaling molecules, forming a dynamic ecosystem that sustains tumor growth through nutrient supply and immunosuppressive reprogramming [25]. To characterize the tumor microenvironment, we employed ESTIMATE, CIBERSORT, ssGSEA, and TIDE to assess immune cell infiltration, functional activity, and immunotherapy response prediction. We found elevated MDSC scores suggesting that combinatorial myeloid-targeted therapy may surpass T cell monotherapy [30]. And prognostic CAF scores indicating stromal targeting could enhance T cell infiltration [31]. Integrated analysis revealed elevated tumor mutational burden and immunosuppressive myeloid in high-risk patients, highlighting combinatorial targeting strategies to overcome therapeutic resistance.

TYMP catalyzes the phosphorolysis of thymidine to thymine and 2-deoxy-D-ribose-1-phosphate [32]. Classified within glycosyltransferase gene sets in MSigDB due to its dual GT family domains, TYMP exhibits both enzymatic and non-enzymatic functions [33]. The N-terminal proline-rich domain engages Src family kinases via SH3 binding, modulating platelet activation and thrombosis [34]. This suggests that structural domain combinatorics underlie multifunctionality, with glycosyltransferase domains providing molecular scaffolds for substrate binding and conformational dynamics. Previous omics studies have found that TYMP is associated with poor prognosis in ccRCC [35,36]. Our analysis found the expression of TYMP was significantly elevated in ccRCC. Notably, TYMP expression correlated positively with immune checkpoint PDCD1 and was elevated in anti-PD-1 responders, supporting its potential as an immunotherapy biomarker.

GCNT4 mediates the formation of core2 in O-GalNAc glycosylation, which stabilizes the normal conformation of cell surface mucins and blocks abnormal glycosylation ligand receptor interactions between tumor cells and immune cells in the microenvironment [37,38]. The complete performance of the core2 structure inhibits the expression of tumor-associated glycoantigens, and truncated core2 structures are often associated with tumor metastasis and immune evasion [39,40]. In ovarian cancer, the maintenance of the core2 structure can also regulate the polarization of tumor-associated macrophages, inhibit their transformation into immunosuppressive phenotypes, and promote the recognition efficiency of T cells towards tumor neoantigens [41,42]. We found that the expression of GCNT4 was significantly downregulated in clear cell renal cell carcinoma. GCNT4 has a significant impact on the prognosis of patients with clear cell renal cell carcinoma, and patients with high expression of GCNT4 have significantly increased survival time.

Our findings provide potential clinical implications for the stratification of patients with ccRCC. The GTRS model enables classification of patients into high- and low-risk subgroups with distinct prognostic outcomes, tumor mutational burdens, immune microenvironment characteristics, and therapeutic sensitivities. Patients in the high-risk group exhibited increased tumor aggressiveness, higher levels of immunosuppressive cell infiltration, and elevated expression of immune checkpoint molecules, suggesting potential resistance to immunotherapy. These patients may benefit from combination regimens involving immune checkpoint inhibitors and tyrosine kinase inhibitors. Conversely, patients in the low-risk group demonstrated a more active immune phenotype and higher predicted sensitivity to certain chemotherapeutic and targeted agents, indicating that they may respond favorably to conventional or less intensive therapeutic approaches. Furthermore, by integrating GTRS with clinical parameters such as age, pathological grade, and tumor stage in a nomogram, our model enhances the precision of prognostic stratification and facilitates personalized treatment decision-making for ccRCC patients.

In summary, our study established a prognostic model for ccRCC by integrating 117 algorithm combinations derived from eight machine learning methods based on a curated set of 337 glycosyltransferase genes. The StepCox[forward] + Ridge algorithm was selected to construct the GTRS based on 13 prognostic glycosyltransferase genes. Further screening revealed TYMP, GCNT4, UGT2B7, HS3ST2, and FUT3 as key genes in the gene model. Among them, TYMP is one of the key poor prognostic genes in the model, and the experiment verified it to be associated with the malignant biological behavior of ccRCC. Overexpression of GCNT4 is associated with good prognosis, and knocking down GCNT4 promotes the malignant phenotype of ccRCC. Our finding shows that the GTRS system provides a clinically actionable tool for risk stratification, offering a mechanistic foundation for precision oncology in ccRCC.

## 4. Materials and Methods

### 4.1. Data Acquisition

Four glycosylation-related databases and datasets were selected to identify glycosyltransferase genes. (1) GlycoGene Database (GGDB): All genes were collected as the most cited resource [43]. (2) HGNC (Group 424): Includes computationally predicted and experimentally verified genes [44]. (3) MSigDB: The “GOMF_GLYCOSYLTRANSFERASE_ACTIVITY” gene set was retrieved [45]. (4) GlycCosmos: Human glycosyltransferase genes from this glycomics database were extracted. Taking the union of identified genes, we had 337 unique glycosyltransferase genes.

Data Acquisition: RNA sequencing datasets from three ccRCC cohorts were analyzed—TCGA (*n* = 437), CPTAC (*n* = 103), and EMTAB1980 (*n* = 101). Multi-omics data for TCGA were downloaded using the R package TCGAbiolinks (v2.32.0), with updated clinical data obtained directly from the GDC portal [46]. Data for the CPTAC cohort (PDC000127) were retrieved from the PDC database. The EMTAB1980 dataset was sourced from ArrayExpress.

### 4.2. Data Normalization

For transcriptomic data across all cohorts, expression values of duplicate mRNAs were averaged. Count data were normalized using the R package limma (v3.60.6) [47]. FPKM/RPKM-formatted data were converted to TPM. Prior to downstream analyses, TPM values underwent log2(x + 1) transformation.

### 4.3. Integrated Machine Learning Model Construction

The GTRS model was developed using the R package Mime1 (v0.0.0.9000) [28]. This framework integrated eight algorithms: LASSO, Ridge, Elastic Net (Enet), Stepwise Cox Regression (StepCox), CoxBoost, plsRcox, Supervised Principal Components (SuperPC), Generalized Boosted Regression Model (GBM), survival-SVM, and Random Survival Forest (RSF). All possible pairwise algorithm combinations (*n* = 117) were evaluated. Models were trained using k-fold cross-validation.

Expression matrices, corresponding survival data from the TCGA, CPTAC, and EMTAB1980 cohorts, and prognostic glycosyltransferase DEGs were input into the integrated machine learning framework. The C-index was calculated for each algorithm within each cohort. The optimal algorithm was selected based on the highest mean C-index across all three datasets. Patients within each cohort were stratified into high-risk and low-risk groups based on the median risk score of their respective training set, applying the same scoring formula uniformly across all datasets.

### 4.4. Cox Regression and Nomogram Construction

Univariate and multivariate Cox regression analyses were performed to assess the independent prognostic value of the model alongside clinical features. The R package rms (v6.8.2) was used to construct a prognostic nomogram based on significant independent factors. Nomogram performance was evaluated using calibration curves. Cox regression analyses were conducted with the survival package, and results were visualized using the ezcox package (v1.0.4).

### 4.5. TMB Analysis

Somatic mutation data from TCGA were analyzed using the R package maftools (v2.20.0). This included assessment of mutation frequency, spectrum, and genomic distribution [48]. TMB was calculated per sample using single-nucleotide variant data and compared between high- and low-risk groups. Mutation landscapes, including summary plots and waterfall plots, were visualized using the plotmafSummary function.

### 4.6. Tumor Immune Microenvironment Analysis

Stromal score, immune score, and tumor purity were calculated using the ESTIMATE algorithm [49]. Infiltrating immune cell proportions were analyzed via the CIBERSORT algorithm [50]. Enrichment levels of specific immune pathways or cell types within each sample were assessed using ssGSEA [51].

### 4.7. Chemotherapy Drug Sensitivity Analysis

Half-maximal inhibitory concentration (IC50) data for common chemotherapeutic and molecularly targeted agents in ccRCC were retrieved from the Genomics of Drug Sensitivity in Cancer (GDSC) database. The R package oncoPredict (v0.2) was used to calculate, visualize, and compare IC50 values between high-risk and low-risk patient groups [52]. Higher IC50 values indicate lower drug sensitivity.

### 4.8. Functional Enrichment Analysis

Gene Ontology (GO) and Kyoto Encyclopedia of Genes and Genomes (KEGG) pathway enrichment analyses were performed using the R package clusterProfiler (v4.12.6) with the enrichGO and enrichKEGG functions, respectively [53]. Gene Set Enrichment Analysis (GSEA) was conducted using the GSEA function within clusterProfiler to identify systemic biological features associated with phenotypes. GSEA results were visualized using gseaplot2 function in enrichplot (v1.16.1). Gene sets with a normalized enrichment score (NES) absolute value > 1 were considered significantly enriched, where positive and negative NES values indicate positive or negative enrichment, respectively.

### 4.9. Cell Line Culture

The 786-O, CAKI-1, and A498 cell lines were cultured in RPMI-1640 medium supplemented with 10% fetal bovine serum (FBS). The HK-2 cell line was maintained in DMEM medium supplemented with 10% FBS. All cells were incubated at 37 °C in a humidified atmosphere containing 5% CO_2_. Medium for 786-O, CAKI-1, and 498 was replaced every other day, while medium for HK-2 was replaced every three days. Cells were passaged and cryopreserved upon reaching 80–90% confluence. All cell lines have been authenticated using STR profiling within the last three years. And all experiments were performed with mycoplasma-free cells.

### 4.10. Cell Counting Kit 8 (CCK-8) Assay and Colony Formation Assay

CCK-8 assay was measured using the Cell Counting Kit (New Cell & Molecular Biotech, China, cat.C6005) according to the manufacturer’s protocol. Briefly, the cells were seeded in 96-well plates at 2 × 10^3^ cells/well and measured at optical density 450 nm using Tecan Infinite F50 (Tecan) every 24 h over 5 days. In colony formation assay, 0.5 × 10^3^ cells were seed in each well of 6-well plates and incubated for 14 days. The colonies were washed twice by PBS and fixed with 4% formaldehyde for 10 min. Then, the colonies were stained with 0.1% crystal violet (Solarbio, China, cat. G1063) for 30 min. The images of colonies were photographed using ChemiDoc MP Imaging System (BioRad, Hercules, CA, USA). ImageJ (v1.53c) was used to count the numbers of colonies in the well.

### 4.11. Transwell Assay,

The invasion and migration assays were performed using transwell inserts (Corning, Los Angeles, CA, USA) with or without Matrigel in 24-well plates according to the manufacturer’s protocol. Briefly, cells were seeded at a density of 5 × 10^4^ cells per upper well in a 200 μL culture medium containing 2% FBS, with the lower chambers containing 500 μL culture medium which contained 20% FBS as a chemoattractant stimulus. The cells were incubated for 24 h or 48 h at 37 °C and allowed to migrate or invade through the membrane filter. Then, the noninvasive and nonmigratory cells fixed to upper well were gently removed using a moist cotton swab. The cells that had invaded or migrated to the bottom surface of the membrane were fixed with 4% formaldehyde for 1 min and stained with hematoxylin and eosin (H&E) for 5 min and 2 min, respectively. Four fields of view were randomly selected for quantification under the microscope (Leica, Wetzlar, Germany).

### 4.12. Western Blot

Western blot was performed according to the standard protocol. Total protein was extracted using prechilled RIPA Buffer (Thermo Fisher Scientific, Waltham, MA, USA, cat.89901) with 1% protease inhibitor cocktails (Bimake, Houston, TX, USA, cat. B14001). Protein extracts were subjected to electrophoresis on 10% SDS-PAGE and transferred onto polyvinylidene fluoride (PVDF) membranes (Millipore, Temecula, CA, USA, cat. ISEQ00010). Immunoblots were blocked with 5% BSA in TBS/Tween20 and incubated separately with antibody overnight at 4 °C. Subsequently, all membranes were incubated with HRP-labeled mouse IgG secondary antibody (CST, Danvers, MO, USA, cat.7076P2) or HRP-labeled rabbit IgG secondary antibody (CST, cat. 7074P2). The NcmECL Ultra kit (New Cell & Molecular Biotech, cat. P10300) was used to visualize the protein bands. The protein bands were photographed by ChemiDoc MP Imaging System (BioRad).

### 4.13. Statistical Analysis

Bioinformatics analyses and visualization were performed using R (v4.4.1). Associations were assessed as follows: Spearman correlation for risk score and immune cell infiltration abundance; Wilcoxon rank-sum test for comparisons between two groups; and Pearson correlation for gene–gene relationships. All wet-lab experiments were performed in triplicate. Experimental data were analyzed using SPSS (v23.0), with visualization in ImageJ (v1.53c) and GraphPad Prism (v9.5.1). Continuous data normality was confirmed via Kolmogorov–Smirnov test. Following confirmation of normality and variance homogeneity (assessed via ANOVA F-test), group differences were evaluated using Student’s *t*-test. A significance threshold of *p* < 0.05 was applied throughout.

## 5. Conclusions

This study systematically defines the prognostic landscape of glycosyltransferases in ccRCC, establishing that glycosyltransferase gene sets critically determine patient outcomes and enable robust predictive modeling. The StepCox[forward] + Ridge algorithm was selected to construct the GTRS based on 13 prognostic glycosyltransferase genes. Core drivers identified include TYMP, GCNT4, UGT2B7, HS3ST2, and FUT3. The experimental validation confirms TYMP as a key adverse prognostic factor promoting malignant phenotypes and GCNT4 overexpression correlating with favorable outcomes, while its knockdown accelerates aggressiveness.

## Figures and Tables

**Figure 1 ijms-26-10182-f001:**
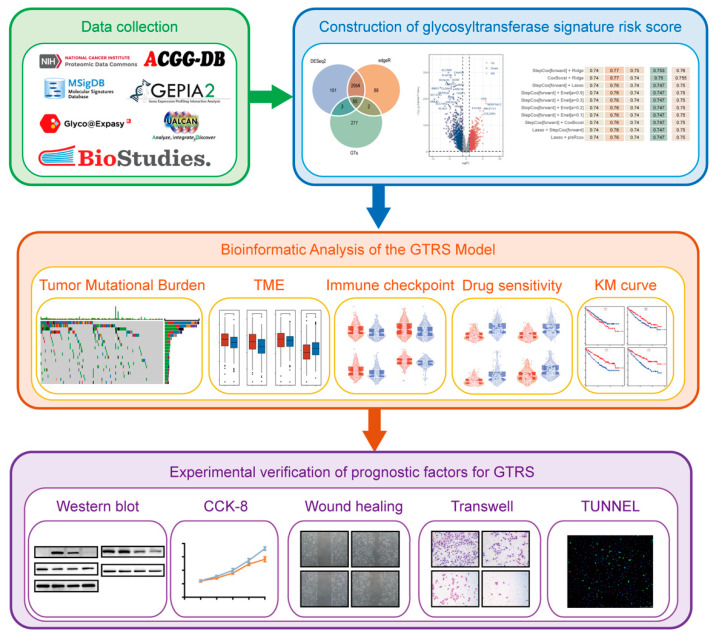
The schematic diagram of the analysis workflow for this study. TME, tumor microenvironment; KM, Kaplan–Meier; TUNNEL, TdT-mediated dUTP Nick-End Labeling.

**Figure 2 ijms-26-10182-f002:**
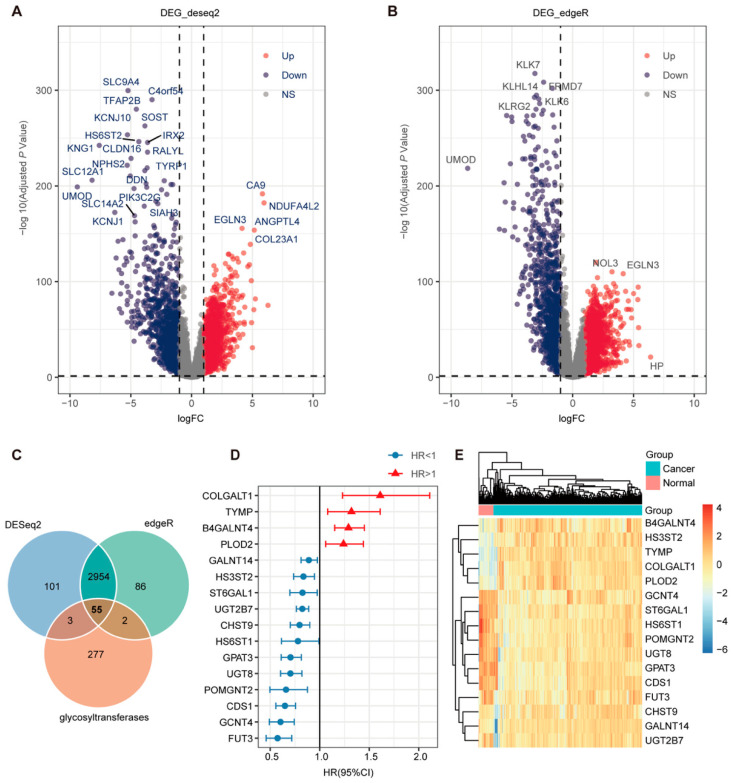
Characterization of glycosyltransferase gene expression and its prognostic significance in ccRCC. (**A**,**B**) Volcano plots showed DESeq2 and edgeR analyses; upregulated genes are marked in red, downregulated genes in blue, and non-significant genes in gray. (**C**) Venn diagram illustrating the overlap of DEGs identified by DESeq2, edgeR, and glycosyltransferase genes. (**D**) Cox regression analysis was used to identify prognostic markers; HR < 1 is indicated by blue circles and HR > 1 by red triangles. (**E**) Heatmap showing the expression profiles of the 16 prognostic glycosyltransferase genes. (DESeq2 and edgeR analysis adjusted *p* < 0.05 and |log2FC| > 1 was defined as having statistical significance.).

**Figure 3 ijms-26-10182-f003:**
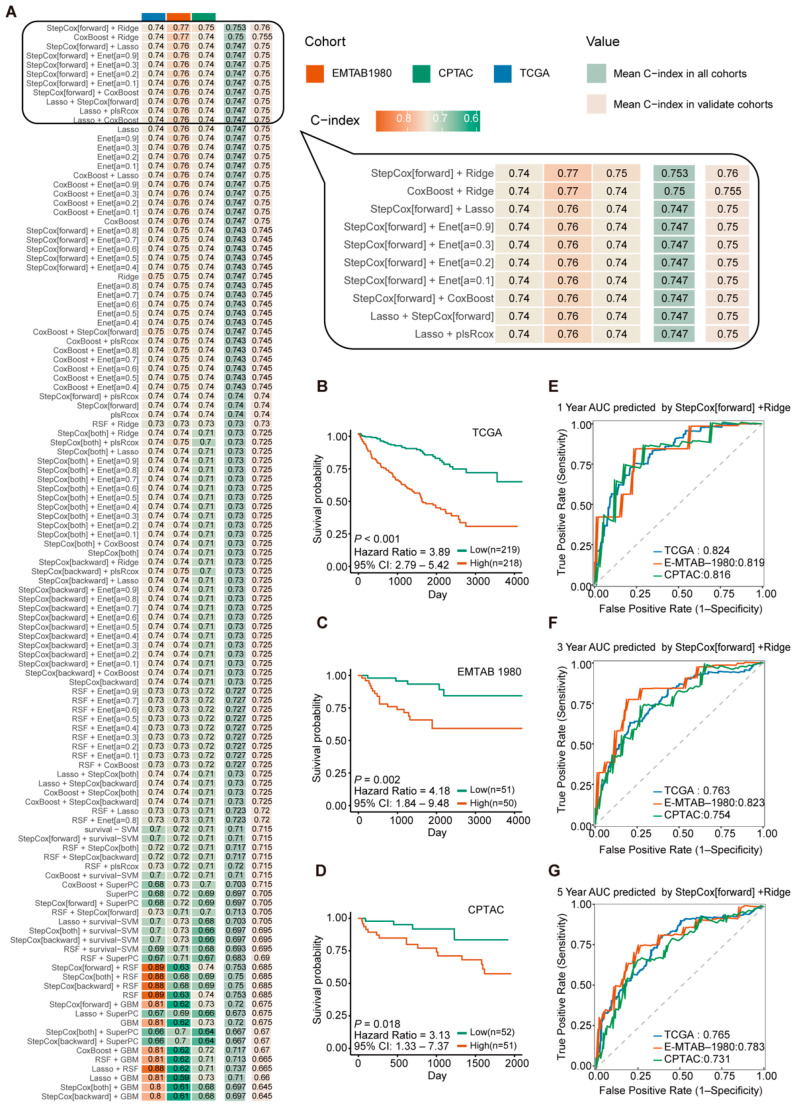
Development and validation of a glycosyltransferase-related gene signature for prognosis in ccRCC. (**A**) show C-indexes of various models in different cohorts, highlighting the top 10 algorithms with the highest C-indexes. (**B**–**D**) show the relationship between risk scores generated by the algorithm with the highest C-index, “Stepcox[forward]+Ridge”, and the survival outcomes of patients in different cohorts. (**E**–**G**) present the 1, 3, and 5-year Receiver Operating Characteristic curves for the three cohorts (log-rank test).

**Figure 4 ijms-26-10182-f004:**
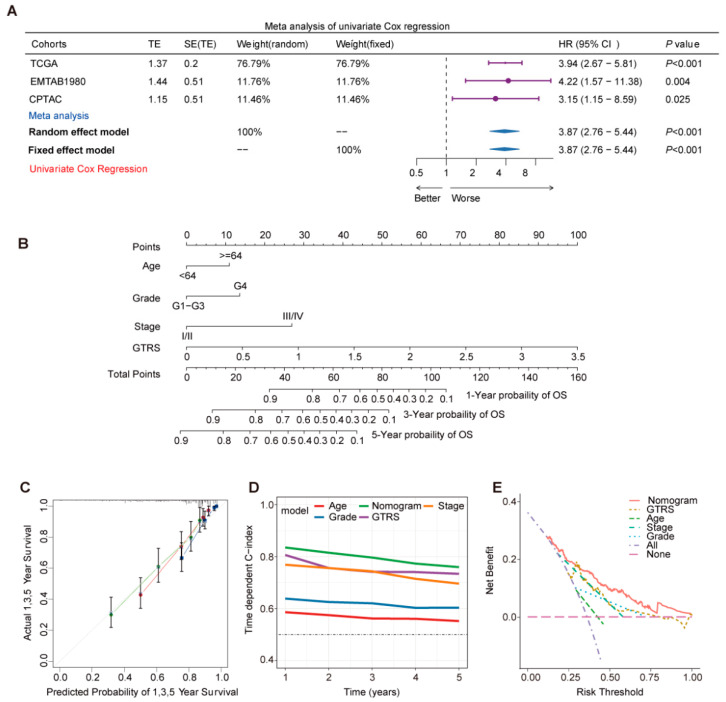
Constructed nomogram to evaluate the predictive ability of the GTRS model. (**A**) Univariate Cox regression analysis shows the risk ratio and 95% confidence interval of each cohort, random effects model, and fixed effects model. (**B**) Nomogram integrating the GTRS, patient age, pathological grade, and clinical stage, illustrating the contribution of each clinical factor to the outcome. The total point score is derived from the sum of individual scores assigned to each variable. The bottom three axes indicate the predicted 1-, 3-, and 5-year survival probabilities corresponding to each total score. (**C**) Calibration curves of the nomogram for predicting 1-, 3-, and 5-year survival. The red, green, and blue lines represent the calibration curves for 1-, 3-, and 5-year survival, respectively. (**D**) Time-dependent C-index curves for each model. (**E**) Decision Curve Analysis evaluating the clinical utility of each model.

**Figure 5 ijms-26-10182-f005:**
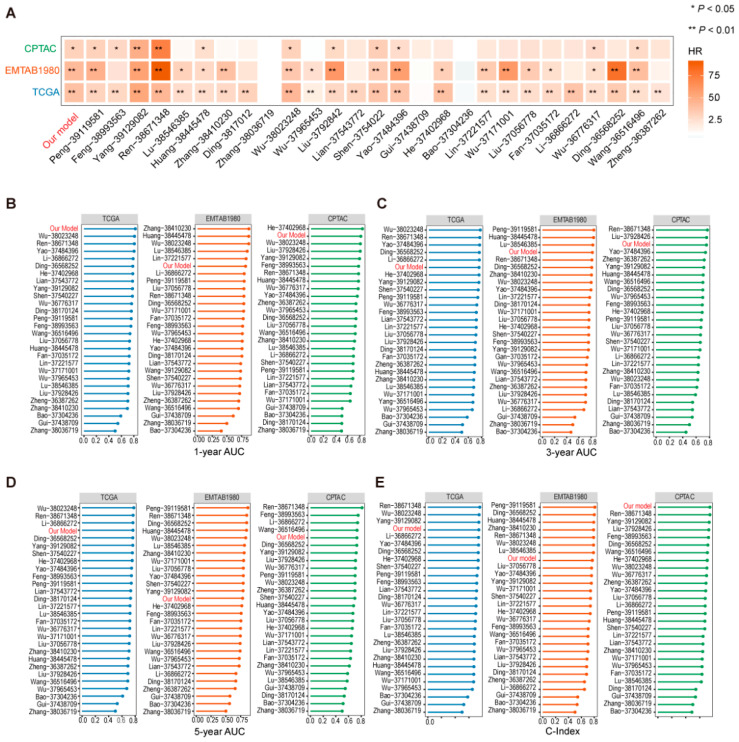
Comparison of the performance between GTRS and other prognostic models. Comparison and analysis of HR values (**A**), 1-year AUC values (**B**), 3-year AUC values (**C**), 5-year AUC values (**D**), and C-index (**E**) between GTRS and 27 other models. (* *p* < 0.05, ** *p* < 0.01, Likelihood Ratio Test).

**Figure 6 ijms-26-10182-f006:**
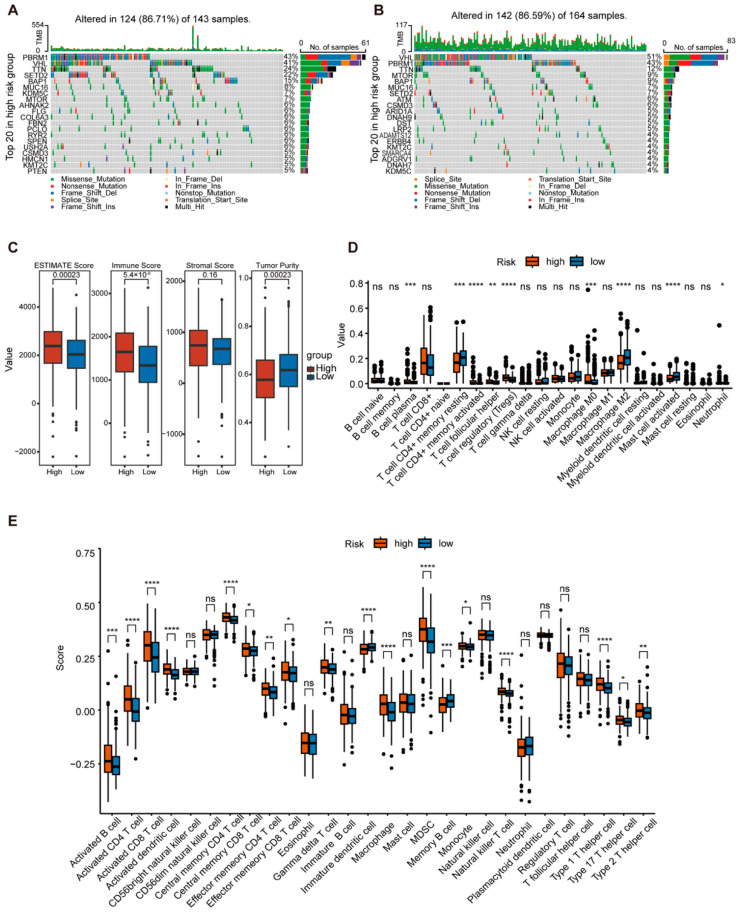
The TMB and tumor microenvironment analysis of GTRS. (**A**,**B**) Mutation maps of the top 20 genes in the high-risk and low-risk groups of GTRS in the TCGA cohort. (**C**) Immune score, ESTIMATE score, stromal score, and tumor purity calculated by ESTIMATE. (**D**) CIBERSORT evaluated the expression of microenvironment cells. (**E**) ssGSEA algorithm evaluates the types of immune cells. (* *p* < 0.05; ** *p* < 0.01; *** *p* < 0.001, **** *p* < 0.0001, ns = no significance; independent Student’s *t*-test).

**Figure 7 ijms-26-10182-f007:**
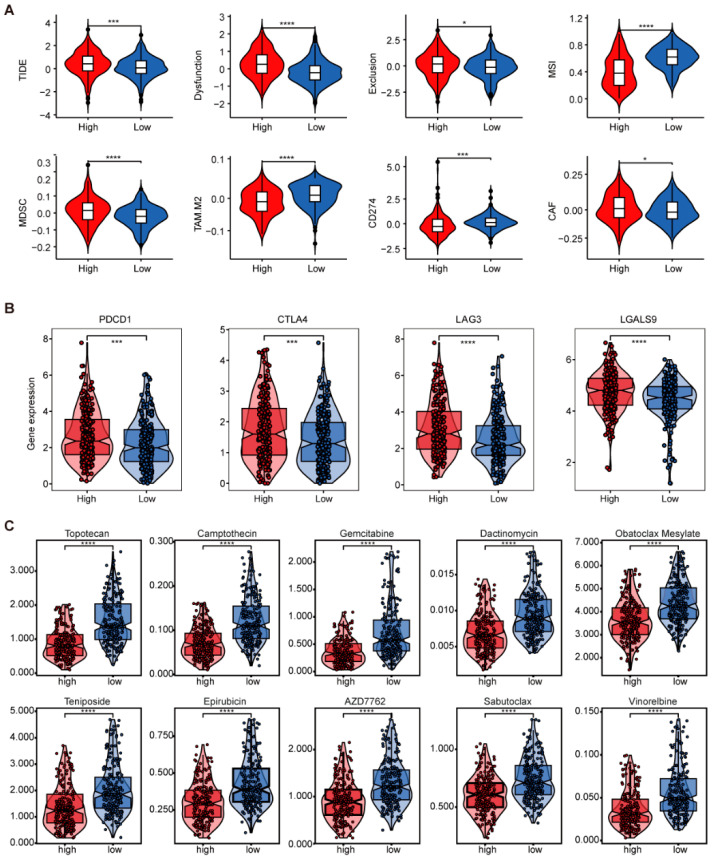
Differences in immune therapy and drug sensitivity between GTRS high-risk and low-risk groups. (**A**) Differences in the scores of TIDE, dysfunction, exclusion, MSI, (E)MDSC, TAM. M2, CD274, and CAF between the high-risk and low-risk groups of GTRS were compared. (**B**) Comparison of the expression levels of immune checkpoint genes PDCD1, CTLA4, LAG3, and LGALS9 between the high-risk and low-risk groups of GTRS. (**C**) Comparison of IC50 values of 10 drugs in GTRS high-risk and low-risk groups. (* *p* < 0.05; *** *p* < 0.001, **** *p* < 0.0001; independent Student’s *t*-test).

**Figure 8 ijms-26-10182-f008:**
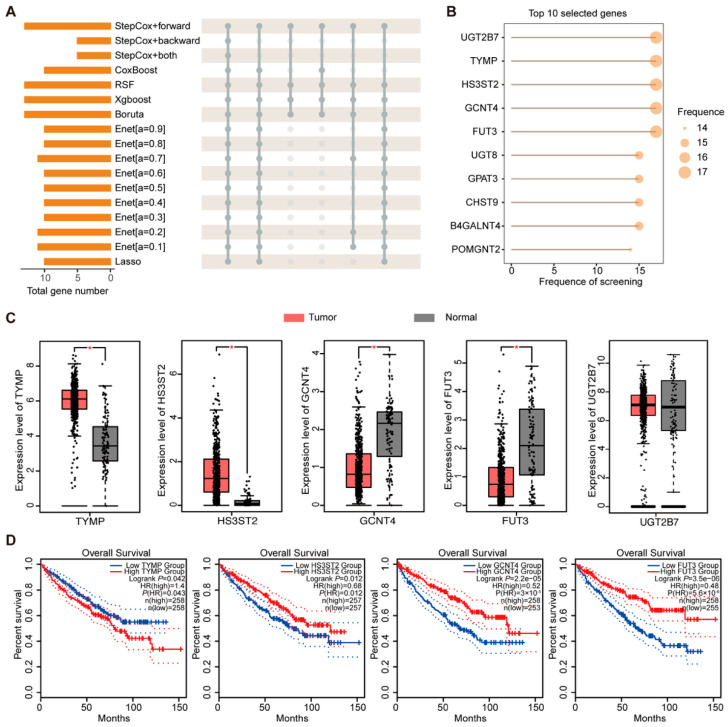
Prognosis-related genes selected by different machine learning algorithms. (**A**) The number of different genes in multiple machine learning models. (**B**) The frequency of genes selected by different machine learning models. (**C**) The box plots showing the expression differences in TYMP, HS3ST2, GCNT4, FUT3, and UGT2B7 between TCGA tumor samples (N = 537) and TCGA non-tumor samples plus GTEx samples (N = 100). (**D**) Kaplan–Meier curves for overall survival in relation to the expression levels of TYMP, HS3ST2, GCNT4, and FUT3. (* *p* < 0.05, log-rank test and independent Student’s *t* test).

**Figure 9 ijms-26-10182-f009:**
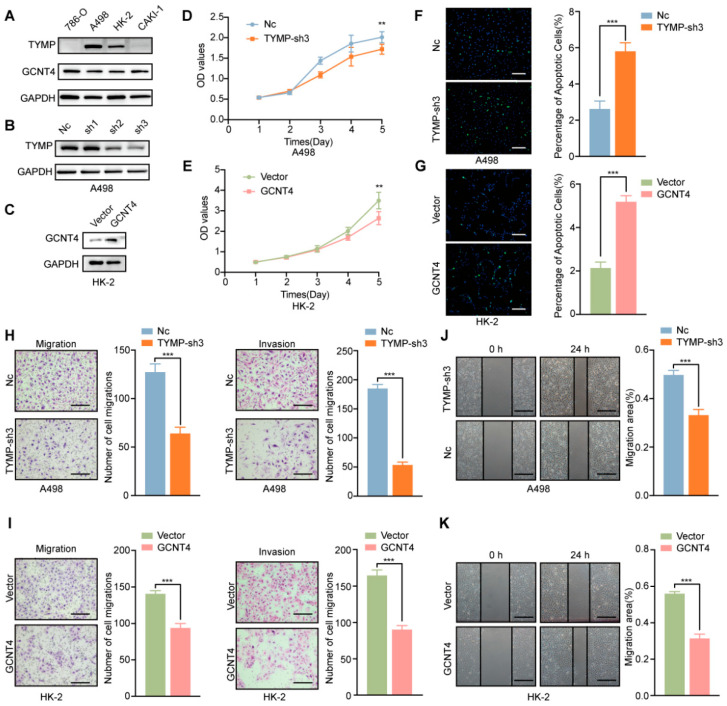
Altering the expression of TYMP and GCNT4 externally affects the proliferation, migration, and invasion of ccRCC. (**A**) Endogenous TYMP and GCNT4 expression in four ccRCC cells. (**B**) Western blot showing knockdown of TYMP in A498 cells. (**C**) Western blot showing GCNT4 ectopic expression in transfected HK-2 cells. (**D**,**E**) CCK8 assay indicating TYMP knockdown and GCNT4 overexpression affected cell growth. (**F**,**G**) Tunnel assay indicated TYMP knockdown and GCNT4 overexpression affected cell apoptotic (bar = 250 µm). (**H**,**I**) transwell migration and invasion assay showing that TYMP knockdown and GCNT4 overexpression affected cell migration and invasion (bar = 100 µm). (**J**,**K**) The wound healing assay showing that TYMP knockdown and GCNT4 overexpression affected cell motility (bar = 100 µm). (** *p* < 0.01; *** *p* < 0.001, independent Student’s *t*-test).

## Data Availability

Data is contained within the article, Appendix A, or references cited.

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
