# Peer review of "Constructing a Prognostic Model for Clear Cell Renal Cell Carcinoma Based on Glycosyltransferase Gene and Verification of Key Gene Identification"

_ijms, 2025, doi:10.3390/ijms262010182_

Round 1

Reviewer 1 Report

Comments and Suggestions for Authors

In this work, Chong Zhou et al. aimed to construct a prognostic model for ccRCC based on glycosyltransferase genes.

Minor points

  1. The authors should add in the Discussion section how these findings would be implemented in the stratification of patients with ccRCC.

  1. The authors identify 16 prognostic glycosyltransferase genes (Figure 2D): 4 genes that they classify as risk genes; and 12 genes that they classify as protective genes. It would be interesting for them to delve deeper into these genes as a potential genetic signature relevant to the prognosis and diagnosis of ccRCC.

Author Response

Thank you for your kind letter and constructive comments concerning our manuscript (ijms-3876272) entitled "Constructing a Prognostic Model for Clear Cell Renal Cell Carcinoma Based on Glycosyltransferase Gene and Verification of Key Gene Identification". We appreciate your help and suggestions. These comments are all valuable and helpful for improving our article as well as research. All the authors have seriously discussed all these comments. Over the past ten days, we have made great efforts to revise the suggestions you put forward to meet the requirements of the journal. In the revised version, changes to our manuscript within the document were all highlighted by using red colored text. Point-by-point responses to the comments are listed below this letter. To make the changes easier to identify we have numbered them.

Comments 1: The authors should add in the Discussion section how these findings would be implemented in the stratification of patients with ccRCC.

Response 1: We appreciate the reviewer’s insightful suggestion. In the revised version, we have added a detailed discussion on how our findings could be applied to the clinical stratification of ccRCC patients. Specifically, our GTRS model constructed from 13 glycosyltransferase genes, was applied clinical to effectively divides ccRCC patients into high and low risk groups with distinct prognostic outcomes, tumor mutational burdens, immune microenvironment characteristics, and predicted responses to immunotherapy and targeted drugs. This addition has been incorporated into the revised Discussion section to clarify the translational potential of our prognostic model for patient stratification and precision therapy (page17, line421).

Comments 2: The authors identify 16 prognostic glycosyltransferase genes (Figure 2D): 4 genes that they classify as risk genes; and 12 genes that they classify as protective genes. It would be interesting for them to delve deeper into these genes as a potential genetic signature relevant to the prognosis and diagnosis of ccRCC.

Response 2: We appreciate the reviewer’s valuable suggestion. In the revised manuscript, a heatmap was added to illustrate their expression patterns between tumor and normal samples. The results showed that the four risk genes were highly expressed in cancer tissues, while the twelve protective genes exhibited higher expression levels in normal tissues, consistent with their prognostic classification (Figure 2E, page 3, line 104). This addition strengthens the biological plausibility of our gene classification and highlights their potential relevance as diagnostic and prognostic biomarkers for ccRCC.

Reviewer 2 Report

Comments and Suggestions for Authors

This study claims to have established a prognostic model for ccRCC by integrating 117 algorithm combinations derived from 8 machine learning methods based on a curated set of 337 glycosyltransferase genes. I have some questions.

1 : Does the pathway of glycosyltransferase genes play a major role in the tumorogenesis and progression of CCRCC ? What is the relationship between the glycosyltransferase genes and the genes of other molecular pathways such as VHL-HIF-CA9 expressions ?  A study of single pathway genes can not reflect all genetic disorders in a tumor. It is difficult to imagine that the prognostic model based only on glycosyltransferase genes can add extra value to the existing models.

2 : The study on prognostic value of GCNT4 was based on the overall survival. The progression-free survival was needed to study its prognostic value. Besides, the duration of followup needed to extend beyond five years. The prognostic value of GCNT4 can not be confirmed.

3 : The abstract is only descriptive. The abstract should be re-written to include concrete results and conclusions.

Author Response

Thank you for your kind letter and constructive comments concerning our manuscript (ijms-3876272) entitled "Constructing a Prognostic Model for Clear Cell Renal Cell Carcinoma Based on Glycosyltransferase Gene and Verification of Key Gene Identification". We appreciate your help and suggestions. These comments are all valuable and helpful for improving our article as well as research. All the authors have seriously discussed all these comments. Over the past ten days, we have made great efforts to revise the suggestions you put forward to meet the requirements of the journal. In the revised version, changes to our manuscript within the document were all highlighted by using red colored text. Point-by-point responses to the comments are listed below this letter. To make the changes easier to identify we have numbered them.

Comments 1: Does the pathway of glycosyltransferase genes play a major role in the tumorogenesis and progression of CCRCC? What is the relationship between the glycosyltransferase genes and the genes of other molecular pathways such as VHL-HIF-CA9 expressions? A study of single pathway genes cannot reflect all genetic disorders in a tumor. It is difficult to imagine that the prognostic model based only on glycosyltransferase genes can add extra value to the existing models.

Response 1: We thank the reviewer for this insightful and constructive comment. In response, we have added relevant content to both the introduction and discussion sections to clarify the functional significance of glycosyltransferase-related pathways in ccRCC tumorigenesis and their interaction with other signaling networks. In the introduction (page 2, line 60), we emphasize that dysregulated glycosylation contributes to ccRCC progression, angiogenesis, and immune modulation, with aberrant expression of specific glycosyltransferases. Furthermore, in the Discussion (page 15, line 342), we have elaborated on the mechanistic link between glycosyltransferases and the VHL–HIF–CA9 pathway. We discuss recent research showing that loss of functional VHL impairs the clearance of misprocessed glycoproteins, while increased O-GlcNAcylation and ST6Gal-I–mediated sialylation can stabilize HIF-1α and enhance glycolytic and hypoxic adaptation. These revisions highlight that glycosyltransferase-related pathways are not isolated mechanisms but are intricately connected with the metabolic and hypoxic signaling cascades central to ccRCC pathogenesis, thereby supporting the biological rationale and added value of our glycosyltransferase-based prognostic model.

Comments 2: The study on prognostic value of GCNT4 was based on the overall survival. The progression-free survival was needed to study its prognostic value. Besides, the duration of followup needed to extend beyond five years. The prognostic value of GCNT4 can not be confirmed.

Response 2: Agree. We have, accordingly, supplemented our analysis by evaluating the disease-free survival to further assess their prognostic value. The results indicated that elevated expression of GCNT4 and FUT3 was associated with longer disease-free survival, supporting their potential protective roles in ccRCC progression. This addition addresses the reviewer’s concern regarding the prognostic evaluation beyond overall survival. The relevant content has been incorporated into the Results section (page 12, line 278)

Comments 3: The abstract is only descriptive. The abstract should be re-written to include concrete results and conclusions.

Response 3: Agree. We have re-written the abstract to include concrete results and conclusions, highlighting the construction and validation of the GTRS model, its prognostic performance, and the experimental validation of key genes. This revision addresses the reviewer’s concern regarding the descriptive nature of the previous abstract. The revised abstract can be found on page 1, line 21 of the manuscript.

Round 2

Reviewer 2 Report

Comments and Suggestions for Authors

The authors have considered all my comments and modified the text accordingly. The article can be accepted.